# INTRUSION-FREE GRAPH MIXUP

## ABSTRACT

We present a simple and yet effective interpolation-based regularization technique to improve the generalization of Graph Neural Networks (GNNs). We leverage the recent advances in Mixup regularizer for vision and text, where random sample pairs and their labels are interpolated to create synthetic samples for training. Unlike images or natural sentences, which embrace a grid or linear sequence format, graphs have arbitrary structure and topology, which play a vital role on the semantic information of a graph. Consequently, even simply deleting or adding one edge from a graph can dramatically change its semantic meanings. This makes interpolating graph inputs very challenging because mixing random graph pairs may naturally create graphs with identical structure but with different labels, causing the manifold intrusion issue. To cope with this obstacle, we propose the first input mixing schema for Mixup on graph. We theoretically prove that our mixing strategy can recover the source graphs from the mixed graph, and guarantees that the mixed graphs are manifold intrusion free. We also empirically show that our method can effectively regularize the graph classification learning, resulting in superior predictive accuracy over popular graph augmentation baselines.

## 1 INTRODUCTION

Graph Neural Networks (GNNs) (Kipf & Welling, 2017; Veličković et al., 2018; Xu et al., 2019) have recently shown its profound successes in many challenging applications, including predicting molecule properties for drug and material discovery (Gilmer et al., 2017; Wu et al., 2018), forecasting protein functions for biological networks (Alvarez & Yan, 2012; Jiang et al., 2017), and estimating circuit functionality in modern circuit design (Zhang et al., 2019). Nevertheless, like other successfully deployed deep networks such as those for image classification (Krizhevsky et al., 2012), speech recognition (Graves et al., 2013) and machine translation (Sutskever et al., 2014; Bahdanau et al., 2014), GNNs are also suffering from the data-hungry issue due to their high modeling freedom. Consequently, researchers have been actively seeking effective regularization techniques for GNNs, aiming to power their learning but to avoid over-smoothing (Li et al., 2018; Wu et al., 2019) and over-fitting (Goodfellow et al., 2016; Zhang et al., 2021) for better model generalization. To this end, data augmentation schemes for regularizing GNNs mostly involve edge and node manipulation (e.g., deletion and addition) on a *single* graph (Rong et al., 2020; Zhou et al., 2020; You et al., 2020).

In this paper, we look into a very successful *pairwise* data augmentation technique for image recognition (Zhang et al., 2018a; Verma et al., 2018; Guo et al., 2019a; Kim et al., 2020) and natural text classification (Guo et al., 2019b; Guo, 2020; Jindal et al., 2020), called Mixup. Mixup was originally introduced by Zhang et al. (2018a) as an interpolation-based regularizer for image classification. It regularizes the learning of deep classification models by training with synthetic samples, which are created by linearly interpolating a pair of randomly selected training samples as well as their modeling targets. Neverthe-less, unlike images or natural sentences, which embrace a grid or linear sequence format, graph

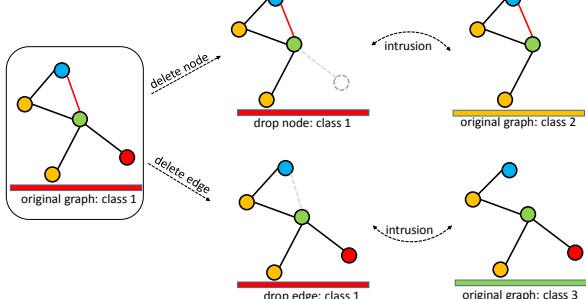

Figure 1: Manifold intrusion caused by deleting a node or an edge (gray dot lines) from the left graph. The two synthetic graphs (middle) have the same structures as the two original graphs on the right but with different labels.

data have arbitrary structure and topology, which play a critical role on the semantics of a graph, and consequently even simply removing or adding one edge can dramatically change the semantic meaning of a graph. As a result, mixing a graph pair is challenging and may naturally cause the manifold intrusion issue in Mixup as identified by Guo et al. (2019a).

Manifold intrusion results from conflicts between the synthetic labels of the mixed-up samples or between the synthetic labels and the labels of the original training data. For example, consider the graph on the left of Figure 1, and its application of the popular graph perturbation action of node and edge deletion (gray lines in the figure). In this case, the resulting two graphs in the middle will have the same structure as the two on the right from the original training set, but with different labels (indicated by the color bars under the graphs). Note: an illustration of manifold intrusion when mixing two graphs is presented in A.5. As discussed in (Guo et al., 2019a), when such intrusions occur, regularization using these synthetic graphs will contradict with the original training data. This essentially induces a form of under-fitting, resulting in the degradation of the model performance.

To address the aforementioned challenge, we propose the first input mixing schema for Mixup on graph learning, coined ifMixup (intrusion-free Mixup). As illustrated in Figure 2, ifMixup first samples a random graph pair (top subfigure) from the training data. For a given mixing ratio $\lambda$ sampled from a Beta distribution, ifMixup then creates a synthetic graph (bottom subfigure), for each sample pair. ifMixup treats a graph pair as two sets of ordered nodes in the same size (middle subfigure, where the unfilled node depicts an added dummy node). As a result, it can linearly interpolate the node features and the edges of the input pair. The newly generated graphs, which have a much larger number of graphs with changing local neighborhood properties than the original training dataset, are then used for training to regularize the GNNs learning. Theoretically, we prove that our mixing strategy can recover the source graphs from the mixed graph, and such invertibility property guarantees that the mixed graphs are intrusion free. That is, our method eliminates the possibility for the graphs resulting from mixing to coincide with any other graph in the training set or with a mixed graph from any other graph pairs. Experimentally, we show, using eight benchmarking tasks from different domains, that our strategy effectively regularizes the graph classification to improve its predictive performance, outperforming popular graph augmentation approaches and existing pair-wise graph mixing methods [1].

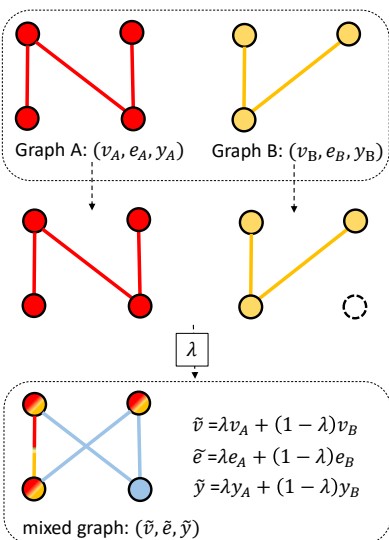

Figure 2: The proposed mixing schema. Top subfigure depicts the source graph pair, and the bottom is the resulting mixed graph for training with mixing ratio $\lambda \in [0, 1]$.

Our key contributions are as follows: 1) To the best of our knowledge, we are the first to propose an input Mixup for graph classification. 2) We prove that our Mixup schema is free of manifold intrusion, which can significantly degrade a Mixup-like model's predictive accuracy. 3) We obtain the SOTA performance among popular graph perturbation baselines.

## 2 RELATED WORK

GNNs have been shown to be very effective for graph classification in a variety of domains (Kipf & Welling, 2017; Ying et al., 2018; Veličković et al., 2018; Klicpera et al., 2019; Xu et al., 2019; Bianchi et al., 2020). One of the key challenges of these successes is to leverage strong regularization techniques such as data augmentation to regularize those GNNs models with high modeling freedom. Nonetheless, data augmentation is still a less touched area in graph data due to the arbitrary structure and topology. Most of the graph augmentation strategies are for node classification tasks, and heavily focus on perturbing nodes and edges in one given graph (Hamilton et al., 2017; Zhang et al., 2018b; Rong et al., 2020; Chen et al., 2020; Zhou et al., 2020; Qiu et al., 2020; You et al., 2020; Wang et al.; Fu et al., 2020; Wang et al., 2020; Song et al., 2021; Zhao et al., 2021). For example, DropEdge (Rong et al., 2020) randomly removes a set of edges of a given graph. GAUG (Zhao et al., 2021) learns to perturb graph edges for node classification. DropNode, representing node

---

[1]Our PyTorch code will be released upon the acceptance of the paper.

sampling based methods (Hamilton et al., 2017; Chen et al., 2018; Huang et al., 2018), samples a set of nodes from a given graph. Unlike these approaches, our proposed strategy leverages a pair of graphs, instead of one graph, to augment the learning of graph level classification.

Despite its great success in augmenting data for image recognition and text processing (Zhang et al., 2018a; Verma et al., 2018; Guo et al., 2019a; Guo, 2020; Jindal et al., 2020; Kim et al., 2020), Mixup has been less explored for graph learning. To the best of our knowledge, there are two methods that apply Mixup to GNNs. GraphMix (Verma et al., 2019) leverages the idea of mixing on the embedding layer, with an additional fully-connected network to share parameters with the GNNs, for graph node classification in semi-supervised learning. MixupGraph (Wang et al., 2021) also leverages a simple way to avoid dealing with the arbitrary structure in the input space for mixing a graph pair, through mixing the graph representation resulting from passing the graph through the GNNs. Our paper here introduces the first input mixing method for Mixup to augment training data for graph classification. Furthermore, our method guarantees that the mixed graphs are manifold intrusion free for Mixup.

## 3 INTRUSION-FREE GRAPH MIXING

### 3.1 GRAPH CLASSIFICATION AND MIXUP INTERPOLATION

**Graph Classification** In the graph classification problem we consider, the input $G$ is a graph labelled with node features, or a "node-featured graph". Specifically, assume that the graph has $n$ nodes, each identified with an integer ID in $[n] := \{1, 2, \ldots, n\}$. The set of edges $E$ of the graph is a collection of unordered pairs $(i, j)$'s of node-IDs, as the current paper considers only undirected graphs (although there is no difficulty to extend the setting to directed graphs). Associated with node $i$, there is a feature vector $v(i)$ of dimension $d$. We will use $v$ to denote the collection of all feature vectors and one may simply regards $v$ as a matrix of dimension $n \times d$. Thus, each input node featured graph $G$ is essentially specified via the pair $(v, E)$. The output $y$ is a class label in finite set $\mathcal{Y} := \{1, 2, \ldots, C\}$, which will be expressed as a (one-hot) probability vector over the label set $\mathcal{Y}$. The classification problem is to find a mapping that predicts the label $y$ for a node-featured graph $G$. The training data $\mathcal{G}$ of this learning task is a collection of such $(G, y)$ pairs .

Modern GNNs use the graph structure and node features to learn a distributed vector to represent a graph. The learning follows the "message passing" mechanism for neighborhood aggregation. It iteratively updates the embedding of a node $h_v$ by aggregating representations/embeddings of its neighbors. The entire graph representation $h_G$ is then obtained through a READOUT function, which aggregates embeddings from all nodes of the graph. Formally, representation $h_i^k$ of node $i$ at the $k$-th layer of a GNN is defined as:

$$h_i^k = \text{AGGREGATE}(h_i^{k-1}, h_j^{k-1} | j \in \mathcal{N}(i), W^k), \tag{1}$$

where $W^k$ denotes the trainable weights at layer $k$, $\mathcal{N}(i)$ denotes the set of all nodes adjacent to $i$, and AGGREGATE is an aggregation function implemented by the specific GNN model (popular ones include Max, Mean, Summation pooling operations), and $h_i^0$ is typically initialized as the input node feature $v(i)$ . The graph representation $h_G$ aggregates node representations $h_v$ using the READOUT graph pooling function:

$$h_G = \text{READOUT}(h_i^k | i \in [n]). \tag{2}$$

This graph representation is then mapped to label $y$ using a standard classification network (for example, a softmax layer).

**Mixup Interpolation** Mixup was introduced by (Zhang et al., 2018a) as an interpolation-based regularizer for image classification. It regularizes the learning of deep classification models by training with synthetic samples, which are created by linearly interpolating a pair of randomly selected training samples as well as their modeling targets. In detail, let $(x_A, y_A)$ and $(x_B, y_B)$ be two training instances, in which $x_A$ and $x_B$ refer to the input images and $y_A$ and $y_B$ refer to their corresponding labels. For a randomly chosen such training pair, Mixup generates a synthetic sample as follows.

$$\widetilde{x} = \lambda x_A + (1 - \lambda) x_B, \tag{3}$$

$$\widetilde{y} = \lambda y_A + (1 - \lambda) y_B, \tag{4}$$

where $\lambda$ is a scalar mixing ratio, sampled from a Beta$(\alpha, \beta)$ distribution with hyper-parameters $\alpha$ and $\beta$. Such synthetic instances $(\widetilde{x}, \widetilde{y})$'s are then used for training.

Motivated by the effectiveness of Mixup in regularizing image classification models, we are naturally motivated to design a similar "Mixup" scheme for graph data, in particular, the node-featured graphs, as are the interest of this paper. When this is possible, we may use the synthetic instances $(\widetilde{G}, \widetilde{y})$'s to learn the modele parameter $\theta$ by minimizing the loss $\mathcal{L}$:

$$\min_{\theta} \mathbb{E}_{(G_A, y_A) \sim \mathcal{G}, (G_B, y_B) \sim \mathcal{G}, \lambda \sim Beta(\alpha, \beta)} \lceil \mathcal{L}(\widetilde{G}, \widetilde{y}) \rceil. \tag{5}$$

To mix $(G_A, y_A)$ and $(G_B, y_B)$, it is straight-forward to apply Equation (4) to obtain the mixed label $\widetilde{y}$. The key question of investigation is how to mix $G_A$ and $G_B$ to obtain $\widetilde{G}$.

It is worth noting that, unlike images or natural sentences which embrace a rigid structure of spatial coordinates or time axis, the underlying "coordinate system" of graph data may have different and arbitrary topology across different instances. Consequently even simply deleting or adding one edge can invalid the semantic meaning of a graph. One simple way to avoid dealing with the arbitrary structure in the input space for mixing a graph pair is to mix their fixed-size graph representation that results from the READOUT function as depicted in Equation 2, namely mixing the two graphs by $\widetilde{G} = \lambda h_{G_A} + (1 - \lambda) h_{G_B}$, as proposed by Wang et al. (2021). We here propose to directly mix the graph inputs with arbitrary sizes for Mixup. Our mixing strategy can recover the source graphs from the mixed graph, and such invertibility guarantees that the mixed graphs are free of manifold intrusion (Guo et al., 2019a), which can cause severe underfitting for Mixup learning.

## 3.2 INVERTIBLE GRAPH MIXING SCHEMA

Now we propose a simple approach, ifMixup, for generating mixed node-featured graph $\widetilde{G}$ from a pair of such graphs $G_A$ and $G_B$. In the nutshell, ifMixup simply adopts a different representation for each node-featured graph.

Specifically, given a node featured graph $G = (v, E)$, we represent $E$ as a binary matrix $e$ with $n$ rows and $n$ columns, in which $e(i, j) = 1$ if $(i, j) \in E$, and $e(i, j) = 0$ otherwise. Thus intead of expressing $G$ as $(v, E)$ we express it as $(v, e)$. The mixing of $G_A = (v_A, e_A)$ with $G_B = (v_B, e_B)$ to obtain $\widetilde{G} = (\widetilde{v}, \widetilde{e})$, can simply be done as follow,

$$\widetilde{e} = \lambda e_A + (1 - \lambda) e_B. \tag{6}$$

$$\widetilde{v} = \lambda v_A + (1 - \lambda) v_B. \tag{7}$$

In order for the above mixing rule to be well defined, we need the two graphs to have the same number of nodes. For this purpose we define $n = \max(n_A, n_B)$, where $n_A$ and $n_B$ are the number of nodes in instances $A$ and $B$ respectively. If $G_A$ or $G_B$ has less than $n$ nodes, we simply introduce dummy node to the graph and make them disconnected from the existing nodes. The feature vectors for the dummy nodes are set to the all-zero vector.

This mixing process is illustrated in Figure 2, where the top subfigure is the source graph pair, and the middle depicts the added dummy node (i.e., the node with unfilled circle). The bottom is the resulting mixed graph, where a mixed color indicates that the resulting node or edge is mixed by existing nodes or edges from both source graphs, and the blue color denotes a node that is mixed with a dummy node or an edge that is mixed with a zero-weighted edge.

It is worth noting that the resulting mixed graphs, through Equations 6 and 7, contain edges with weights between [0, 1]. As a result, during training, this will require the GNN networks be able to take the edge weights into account for message passing. Below we will discuss how the two popular GNN networks, namely GCN (Kipf & Welling, 2017) and GIN (Xu et al., 2019), cope with the weighted edges in graphs, namely how they implement Equation 1 to generate node representations.

GCN handles edge weights naturally by enabling adjacency matrix to have values between zero and one (Kipf & Welling, 2017), instead of either zero or one, representing edge weights:

$$\mathbf{h}_i^k = \sigma \left( W^k \cdot \left( \sum_{j \in \mathcal{N}(i) \cup \{i\}} \frac{e(i, j)}{\sqrt{\hat{d}_j \hat{d}_i}} \mathbf{h}_j^{k-1} \right) \right), \tag{8}$$

where $\hat{d}_i = 1 + \sum_{j \in \mathcal{N}(i)} e(i,j)$; $W^k$ stands for the trainable weights at layer $k$; $\sigma$ denotes the non-linearity transformation, i.e. the ReLu function.

To enable GIN to handle soft edge weight, we replace the sum operation of the isomorphism operator in GIN with a weighted sum calculation. That is, the GIN updates node representations as:

$$\mathbf{h}_i^k = \text{MLP}^k \left( (1 + \epsilon^k) \cdot \mathbf{h}_i^{k-1} + \sum_{j \in \mathcal{N}(i)} e(i,j) \cdot \mathbf{h}_j^{k-1} \right), \tag{9}$$

where $\epsilon^k$ is a learnable parameter.

### 3.3 INVERTIBILITY AND INTRUSION-FREENESS

We now show that such a simple mixing scheme in fact makes the original two node-featured graphs $G_A$ and $G_B$ recoverable from the mixed graph $\widetilde{G}$ under a mild assumption and hence avoids manifold intrusion.

To see this, we first show that the graph topology and node features of the two original instances can both be recovered from the mixed instance.

**Lemma 1 (Edge Invertibility)** *Let $\widetilde{e}$ be constructed using Equation 6 with $\lambda \neq 0.5$. Consider equation*

$$se + (1-s)e' = \widetilde{e}$$

*with unknowns $s$, $e$ and $e'$, where $s$ is a scalar and $e$ and $e'$ are binary (i.e., $\{0,1\}$-valued) $n \times n$ matrices. There are exactly two solutions to this equation:*

$$\begin{cases} s = \lambda, & e = e_A, & e' = e_B, \text{or} \\ s = 1 - \lambda, & e = e_B, & e' = e_A \end{cases}$$

Note: the proof of this lemma is in Section A.1.

By this lemma, we see that if the mixing coefficient $\lambda \neq 0.5$, from the mixed edge representation $\widetilde{e}$, we can always recover $e_A$ and $e_B$ (and hence $E_A$ and $E_B$) and their corresponding weights used for mixing. Note that if $\lambda$ is drawn from a continuous distribution over $(0,1)$, the probability it takes value $0.5$ is zero. That is, the connectivity of original two graphs can be perfectly recovered from the mixed edge representation $\widetilde{e}$.

**Lemma 2 (Node Feature Invertibility)** *Suppose that the node feature vectors for all instances in the task take values from a finite set $V \subset \mathbb{R}^d$ and that $V$ is linearly independent. Let $\widetilde{v}$ be cosntructed using Equation 7. Let $V^* = V \cup \{\mathbf{0}\}$, where $\mathbf{0}$ denotes the zero vector in $\mathbb{R}^d$. Consider equation*

$$\widetilde{v} = sv + (1-s)v'$$

*in which $n \times d$ matricess $v$ and $v'$ are unknowns with rows taking value in $V^*$. For any fixed $s \in (0,1)$, there is exactly one solution of $(v, v')$ for this equation.*

Note: the proof of this lemma is in Section A.2.

The node feature invertibility in this lemma requires that the node feature vectors are linear independent. Note that if the feature dimension $d$ is larger than the size $|V|$ of $V$ and for each vector in $V$, its elements are drawn at random, then the linear independence property of $V$ is satisfied with high probability. Thus, if we have the modeling freedom in designing the dimension $d$ of the feature vectors (for example, in designing the embedding dimension), choosing a large $d$ will make the linear independence condition of $V$ satisfied. There are however cases in which feature vectors are given and $d < |V|$. In this case, we establish the following result which requires a much weaker condition.

To that end, suppose that the span $\text{SPAN}(V)$ of $V$ is an $m-$dimensional space and $m < d$. Let $B$ be an $m \times d$ matrix whose rows form a basis of $\text{SPAN}(V)$. Any node feature matrix $v$ can then be expressed as

$$v = TB$$

for some matrix $T$ with size $n \times m$. In this case, we may identify a node-featured graph as $(TB, e)$. Let $\mathcal{T}$ denote the collection of all $T$ matrices for all instances in the training set. That is,

$$\mathcal{T} := \{T : (TB, e) \in \mathcal{G}\}$$

**Lemma 3 (Node Feature Invertibility)** *Let $\widetilde{v}$ be constructed using Equation 7 and suppose that $\mathcal{T}$ is linearly independent. Consider equation*

$$\widetilde{v} = sv + (1-s)v'$$

*in which $n \times d$ matrices $v$ and $v'$ are unknowns where $v = TB$ and $v' = T'B$ for some $T$ and $T'$ in $\mathcal{T}$. For any fixed $s \in (0,1)$, there is exactly one solution of $(v, v')$ for this equation.*

Note: the proof of this lemma is in Section A.3.

Note that since each $T$ has size $n \times m$, usually a large number, it is much easier for the linear independence condition of $\mathcal{T}$ to get satisfied in practice.

**Theorem 1 (Intrusion-Freeness)** *Suppose that $\lambda \neq 1/2$ and that either the condition for Lemma 2 is satisfied or the condition for Lemma 3 is satisfied. Then for any mixed node-featured graph $\widetilde{G} = (\widetilde{v}, \widetilde{e})$ constructed using Equations 6 and 7, the two original node-feature graph $G_A$ and $G_B$ can be uniquely recovered.*

*Proof:* Since $\lambda \neq 0.5$, we can recover $e_A$, $e_B$ and $\lambda$ from $\widetilde{e}$. Given $\lambda$ and either the condition for Lemma 2 or the condition for Lemma 3, we can recover $v_A$ and $v_B$ from $\widetilde{v}$. □

By this theorem, there is no other pair $(G'_A, G'_B)$ from the training set $\mathcal{C}$ that can be mixed into $\widetilde{G}$ using any $\lambda$. Thus, manifold intrusion does not occur under the mild condition of the theorem.

We note the intrusion-freeness of the proposed ifMixup scheme relies on the fact that input graphs do not have soft (weighted) edges. We believe however that there is a potential to extend the scheme to graphs with weighted edges. Promising directions include a combination of the following techniques. First, instead of recovering edges and node features in tandem (as shown in the proof Theorem 1), we may consider jointly recover both. Second we may quantize the edge weights to a set of discrete values and consider a judiciously designed distribution for the mixing coefficient $\lambda$. Third, we may insist the ordering of nodes in a graph to reflect certain semantics or graph topology of instance, whereby only allowing a restricted family of alignment schemes of the two graphs before mixing them. It is our interest to investigate in these directions further.

## 4 EXPERIMENTS

### 4.1 SETTINGS

**Datasets** We evaluate our method with eight graph classification tasks from the graph benchmark datasets collection TUDatasets (Morris et al., 2020): PTC_MR, NCI109, NCI1, and MUTAG for small molecule classification, ENZYMES and PROTEINS for protein categorization, and IMDB-M and IMDB-B for social networks classification. These datasets have been widely used for benchmarking such as in Xu et al. (2019) and can be downloaded directly using PyTorch Geometric (Fey & Lenssen, 2019)'s build-in function online [2]. The social networks datasets IMDB-M and IMDB-B have no node features, and we use the node degrees as feature as in (Xu et al., 2019). Data statistics of these datasets are shown in Table 3, including the number graphs, the average node number per graph, the average edge number per graph, the number of node features, and the number of classes.

**Comparison Baselines** We compare our method with four baselines: MixupGraph (Wang et al., 2021), DropEdge (Rong et al., 2020), DropNode (Hamilton et al., 2017; Chen et al., 2018; Huang et al., 2018), and Baseline. For the Baseline model, we use two popular GNNs network architectures: GCN (Kipf & Welling, 2017) and GIN (Xu et al., 2019).

MixupGraph is the only available approach for applying Mixup on graph classification. It leverages a simple way to avoid dealing with the arbitrary structure for mixing a graph pair, through mixing the entire graph representation resulting from the READOUT function of the GNNs. DropEdge and DropNode are two widely used graph perturbation strategies for graph augmentation. DropEdge randomly removes a set of existing edges from a given graph. DropNode randomly deletes a portion of nodes and their connected edges.

GCN and GIN are two popular GNN architectures and have been widely adopted for graph classification. GCN leverages spectral-based convolutional operation to learn spectral features of graph

---

[2]https://chrsmrrs.github.io/datasets/docs/datasets

through aggregation, benefiting from a normalized adjacency matrix, while GIN leverages the nodes' spatial relations to aggregate neighborhood features, representing the state-of-the-art GNN network architecture. We use their implementations in the PyTorch Geometric platform [3]. Note that, for the GCN, we use the GCN with Skip Connection (He et al., 2016) as that in (Li et al., 2019), This Skip Connection powers the GCN to benefit from deeper layers in GNN networks.

**Detail Settings** We follow the evaluation protocol and hyperparameters search of GIN (Xu et al., 2019) and DropEdge (Rong et al., 2020). We evaluate the models using 10-fold cross validation, and report the mean and standard deviation of three runs on a NVidia V100 GPU with 32 GB memory. Each fold is trained with 350 epochs with AdamW optimizer (Kingma & Ba, 2015), and the initial learning rate is reduced by half every 50 epochs. The hyper-parameters we search for all models on each dataset are as follows: (1) initial learning rate $\in \{0.01, 0.0005\}$; (2) hidden unit of size $\in \{64, 128\}$; (3) batch size $\in \{32, 128\}$; (4) dropout ratio after the dense layer $\in \{0, 0.5\}$; (5) DropNode and DropEdge drop ratio $\in \{20\%, 40\%\}$; (6) number of layers in GNNs $\in \{5, 8\}$; (7) Beta distribution for ifMixup, MixupGraph and Manifold Mixup $\in \{Beta(1, 1), Beta(2, 2), Beta(20, 1)\}$. Following GIN (Xu et al., 2019) and DropEdge (Rong et al., 2020), we report the case giving the best 10-fold average cross-validation accuracy.

## 4.2 RESULTS OF USING GCN AS BASELINE

The accuracy obtained by the GCN (Kipf & Welling, 2017) baseline, ifMixup, MixupGraph, DropEdge, and DropNode with GCN on the eight datasets are presented in Table 1 (best results in **Bold**).

|          | GCN Baseline | ifMixup | MixupGraph | DropEdge | DropNode | Rel. Impr. |
|----------|--------------|---------|------------|----------|----------|------------|
| PTC_MR   | 0.621±0.018 | **0.654±0.003** | 0.633±0.012 | 0.653±0.007 | 0.648±0.018 | 5.31% |
| NCI109   | 0.803±0.001 | **0.820±0.005** | 0.801±0.005 | 0.801±0.001 | 0.793±0.015 | 2.12% |
| NCI1     | 0.804±0.005 | **0.819±0.004** | 0.808±0.004 | 0.811±0.002 | 0.805±0.019 | 1.87% |
| MUTAG    | 0.850±0.011 | **0.879±0.003** | 0.860±0.006 | 0.855±0.008 | 0.829±0.006 | 3.41% |
| ENZYMES  | 0.541±0.001 | **0.570±0.014** | 0.551±0.016 | 0.566±0.006 | 0.532±0.006 | 5.36% |
| PROTEINS | 0.742±0.003 | **0.753±0.008** | 0.742±0.003 | 0.750±0.003 | 0.748±0.001 | 1.48% |
| IMDB-M   | 0.515±0.002 | **0.523±0.004** | 0.513±0.003 | 0.514±0.00. | 0.512±0.003 | 1.55% |
| IMDB-B   | 0.758±0.004 | **0.763±0.003** | 0.759±0.002 | 0.762±0.004 | 0.761±0.005 | 0.66% |

Table 1: Accuracy of the testing methods with GCN networks as baseline. We report mean accuracy over 3 runs of 10-fold cross validation with standard deviations (denoted ±). The relative improvement of ifMixup over the baseline GCN is provided in the last row of the table. Best results are in **Bold**.

Results in Table 1 show that ifMixup outperformed all the four comparison models against all the eight datasets. For example, when comparing with the GCN baseline, ifMixup obtained a relative accuracy improvement of 5.36%, 5.31%, and 3.41% on the ENZYMES, PTC_MR, and MUTAG datasets, respectively. When considering the comparison with the Mixup-like approach MixupGraph, ifMixup also obtained superior accuracy on all the eight datasets. For example, ifMixup was able to improve the accuracy over MixupGraph from 80.1%, 80.8%, 63.3%, and 51.3% to 82.0%, 81.9%, 65.4%, and 52.3% on the NCI109, NCI1, PTC_MR and IMDB-M datasets, respectively.

Furthermore, as shown Table 1, unlike all the other augmentation methods (namely MixupGraph, DropEdge, and DropNode), which can degrade the predictive performance of the baseline models, our method never degraded the baseline models' predictive accuracy.

### 4.2.1 MANIFOLD INTRUSION: MIXING RATIOS FOR GRAPH PAIRS

In this ablation study, we evaluate the sensitivity of the graph mixing ratio to the two Mixup-like approaches: ifMixup and MixupGraph. We present the accuracy obtained by these two methods with Beta distribution as Beta(1, 1), Beta(2, 2), Beta(5, 1), Beta(10, 1) and Beta(20, 1) on the first six datasets of Table 3. Results are presented in Figure 4. Note: the density functions of these five Beta distributions are depicted in Figure 3.

Results in Figure 4 show that both MixupGraph and ifMixup obtained superior results on the six testing datasets with Beta(20, 1). Nevertheless, MixupGraph seemed to very sensitive to the mixing ratio distribution. For example, when Beta distributions were (1, 1)

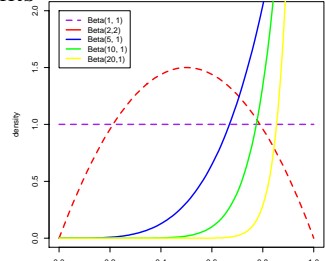

Figure 3: probability density function of Beta distribution.

---

[3]https://github.com/pyg-team/pytorch_geometric

and (2, 2) (first two bars in Figure 4), MixupGraph significantly degraded its accuracy on all the six tasks (except for PTC_MR). In contrast, ifMixup was robust to the five Beta distributions we tested.

We here conjecture that, the decrease of MixupGraph's accuracy obtained with Beta(1, 1) and Beta(2, 2) was due to the manifold intrusion issue. The mixing ratios sampled from Beta(1, 1) follow an uniform distribution between [0, 1], and those sampled from Beta(2, 2) follow a Bell-Shaped distribution between [0, 1] (see Figure 3). Those mixing ratios have a wide range, and thus may aggravate

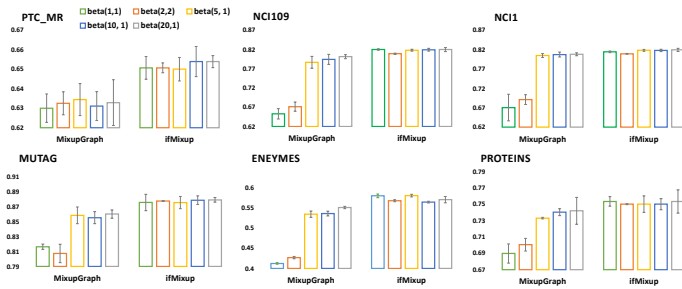

Figure 4: MixupGraph and ifMixup with mixing ratios from Beta(1, 1), Beta(2, 2), Beta(5, 1), Beta(10, 1) and Beta(20, 1).

the creation of mixed embeddings with conflict labels for MixupGraph. On the other hand, ratios being sampled from Beta(5, 1), Beta(10, 1) and Beta(20, 1) mostly fall in the range of [0.8, 1]. Such conservative mixing ratios may alleviate creating conflict training samples for MixupGraph. Promisingly, due to the intrusion-free nature, ifMixup did not suffer from the manifold intrusion problem, showing less sensitivity to the mixing ratios as in Figure 4.

### 4.2.2 OVER-SMOOTHING: IMPACT OF GNNS LAYERS

In this ablation study, we also evaluate the accuracy obtained by GCN, ifMixup and MixupGraph on the first six datasets of Table 3, when varying the number of layers of the GCN networks.

The results for all the six datasets are depicted in Figure 5. Results in this figure show that when increasing the GCN networks from 5 layers (blue bars) to 8 layers (red bars), both GCN and MixupGraph seemed to degrade its performance on all the six datasets. For example, for the NCI109 and NCI1 datasets, MixupGraph resulted in about 10% of accuracy drop when increasing the number of layers in GCNs (with Skip Connection) from 5 to 8. On the contrary, ifMixup was able to increase the accuracy on all the six test datasets.

Such decrease of accuracy obtained by GCN and MixupGraph may due to the over-smoothing problem (Li et al., 2018; Wu et al., 2019). That is, with deeper networks architectures (i.e., more layers), the representations of all nodes of a graph may converge to a subspace that makes these representations unrelated to the input. This negative effect is mainly caused by the fact that the mes-

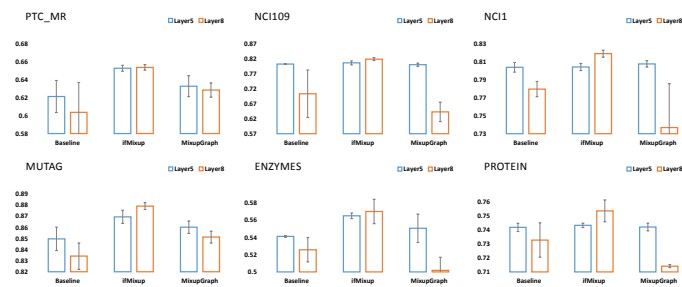

Figure 5: Varying the depth for GCN, ifMixup, and MixupGraph.

sage passing between adjacent nodes is conducted along edge paths in GCNs. That is, each graph convolutional layer in the GCNs keeps pushing the representations of adjacent nodes to blend with each other, based on the fixed connections between nodes. Through generating new adjacency matrices for each training step by randomly deleting a portion of edges of the input graphs, DropEdge (Rong et al., 2020) has show its effectiveness on mitigating over-smoothing. Similar to DropEdge, ifMixup also creates graphs with changing node connections as inputs to the GCNs in each training step. That is, each mixed graph in ifMixup will provide a new adjacency matrix to the networks, making node connections very random and diverse as that in DropEdge. These changing local neighborhood properties in the mixed graphs thus help ifMixup alleviate the over-smoothing problem when GCNs goes deeper.

### 4.2.3 OVER-FITTING: REGULARIZATION EFFECT

In this ablation study, we evaluate the over-fitting effect of our method. We plot the training loss and validation accuracy of ifMixup, GCN, and MixupGraph methods across the 350 training epochs on both the NCI109 and NCI1 datasets in Figure 6.

Figure 6 shows that the training loss of ifMixup (left subfigures) maintained a relatively high level, when compared to GCN and MixupGraph. For GCN, the training loss reduced to near zero after 300 training epochs. Compared to GCN, MixupGraph and ifMixup remained higher training loss, although the loss of MixupGraph kept decreasing after 300 epochs.

The relative high loss here allows the models to keep tuning. Such high loss is due to the much larger space of the synthetic graphs as random and diverse inputs to the networks, thus preventing the model from

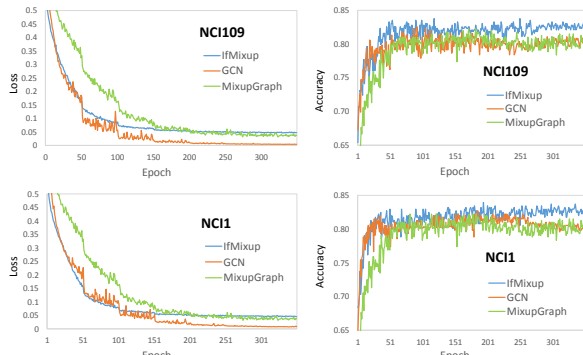

Figure 6: Training loss (left) and validation accuracy (right).

being over-fitted by limited number of graph samples in the original training set. As a result, as shown in the right subfigure, even training for a long time, the ifMixup models were not overfitting.

Note: a 2D visualization of the learned representations of the training graphs is presented in A.6.

## 4.3 Results of using GIN as baseline

We also evaluate our method using the GIN (Xu et al., 2019) network architecture. The accuracy obtained by the GIN baseline, ifMixup, MixupGraph, DropEdge, and DropNode using GIN as baseline on the eight test datasets are presented in Table 2, where best results are in **Bold**.

|  | GIN Baseline | ifMixup | MixupGraph | DropEdge | DropNode | Rel. Impr. |
|---|---|---|---|---|---|---|
| PTC_MR | 0.644±0.007 | **0.672±0.005** | 0.631±0.005 | 0.669±0.003 | 0.663±0.006 | 4.35% |
| NCI109 | 0.820±0.002 | **0.837±0.004** | 0.822±0.008 | 0.792±0.002 | 0.796±0.002 | 2.07% |
| NCI1 | 0.818±0.009 | **0.839±0.004** | 0.822±0.001 | 0.791±0.005 | 0.785±0.003 | 2.57% |
| MUTAG | 0.886±0.011 | **0.890±0.006** | 0.884±0.009 | 0.854±0.003 | 0.859±0.003 | 0.45% |
| ENZYMES | 0.526±0.014 | **0.543±0.005** | 0.521±0.007 | 0.488±0.015 | 0.528±0.002 | 3.23% |
| PROTEINS | 0.745±0.003 | **0.754±0.002** | 0.744±0.005 | 0.749±0.002 | 0.751±0.005 | 1.21% |
| IMDB-M | 0.519±0.001 | **0.532±0.001** | 0.518±0.004 | 0.517±0.003 | 0.516±0.002 | 2.50% |
| IMDB-B | 0.762±0.004 | **0.765±0.005** | 0.761±0.001 | 0.762±0.005 | 0.764±0.006 | 0.39% |

Table 2: Accuracy of the testing methods with GIN networks as baseline. We report mean scores over 3 runs of 10-fold cross validation with standard deviations (denoted ±). The relative improvement of ifMixup over the baseline GIN is provided in the last row of the table. Best results are in **Bold**.

Results in Table 2 show that, similar to the GCN case, the ifMixup with GIN as baseline outperformed all the four comparison models against all the eight datasets. For example, when comparing with GIN, ifMixup obtained a relative accuracy improvement of 4.35%, 3.23%, and 2.57% on the PTC_MR, ENZYMES, and NCI1 datasets, respectively. When comparing with the Mixup-like approach MixupGraph, ifMixup also obtained higher accuracy on all the eight datasets. For example, ifMixup was able to improve the accuracy over MixupGraph from 82.2%, 82.2%, 63.1%, and 51.8% to 83.7%, 83.9%, 67.2%, and 53.2% on the NCI109, NCI1, PTC_MR, and IMDB-M datasets, respectively.

## 5 Conclusions and Future Work

We proposed the first input mixing schema for Mixup on graph classification. We proved that our mixing strategy can recover the source graphs from the mixed graph, and such invertibility in turn guarantees that the mixed graphs are free of manifold intrusion, a form of under-fitting which can significantly degrade a Mixup-like model's predictive accuracy. We showed, using eight benchmark graph classification tasks from different domains, that our strategy obtained superior predictive accuracy over popular graph augmentation approaches and existing pair-wise graph mixing methods.

In the future, we will extend our method for node classification in graph. Also, we will study the potential of our graph mixing schema on semantic-persevering graph mutation.

## 6 REPRODUCIBILITY STATEMENT

Our results are easily reproducible by following the experimental settings in Section 4.1 since our implementations used the PyTorch Geometric platform and the standard datasets from TUDatasets.

Also, we will make our PyTorch code publicly available upon the acceptance of the paper.

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

# A Appendix

## A.1 Proof of Lemma 1

*Proof:* First note that the values in matrix $\widetilde{e}$ can only take values in $\{0, \lambda, 1 - \lambda, 1\}$.

The set $[n] \times [n]$ of all node pairs can be partitioned into four sets:

$$
\begin{aligned}
\mathcal{M}_{00} &:= \{(i,j) \in [n] \times [n] : e_A(i,j) = 0, e_B(i,j) = 0\} \\
\mathcal{M}_{01} &:= \{(i,j) \in [n] \times [n] : e_A(i,j) = 0, e_B(i,j) = 1\} \\
\mathcal{M}_{10} &:= \{(i,j) \in [n] \times [n] : e_A(i,j) = 1, e_B(i,j) = 0\} \\
\mathcal{M}_{11} &:= \{(i,j) \in [n] \times [n] : e_A(i,j) = 1, e_B(i,j) = 1\}
\end{aligned}
$$

It is clear that

$$
\widetilde{e}(i,j) = \left\{
\begin{array}{ll}
0, & \text{if } (i,j) \in \mathcal{M}_{00} \\
1 - \lambda, & \text{if } (i,j) \in \mathcal{M}_{01} \\
\lambda, & \text{if } (i,j) \in \mathcal{M}_{10} \\
1, & \text{if } (i,j) \in \mathcal{M}_{11}
\end{array}
\right.
$$

Let $e, e'$ and $s$ be the solution of the equation in the lemma. On $\mathcal{M}_{00} \cup \mathcal{M}_{11}$, we must have $e = e' = \widetilde{e}$. We only need to determine $e$ and $e'$ on $\mathcal{M}_{01}$ and $\mathcal{M}_{10}$. When $\lambda \neq 0.5$, we must have either

$$
s = \lambda, \ e(i,j) = \left\{
\begin{array}{ll}
1, & (i,j) \in \mathcal{M}_{10} \\
0, & (i,j) \in \mathcal{M}_{01}
\end{array}
\right.
\text{ and } e'(i,j) = \left\{
\begin{array}{ll}
0, & (i,j) \in \mathcal{M}_{10} \\
1, & (i,j) \in \mathcal{M}_{01}
\end{array}
\right.
$$

or

$$
s = 1 - \lambda, \ e(i,j) = \left\{
\begin{array}{ll}
0, & (i,j) \in \mathcal{M}_{10} \\
1, & (i,j) \in \mathcal{M}_{01}
\end{array}
\right.
\text{ and } e'(i,j) = \left\{
\begin{array}{ll}
1, & (i,j) \in \mathcal{M}_{10} \\
0, & (i,j) \in \mathcal{M}_{01}
\end{array}
\right.
$$

Comparing such solutions with $e_A$ and $e_B$, we prove the lemma. $\square$

## A.2 Proof of Lemma 2

*Proof:* We will prove the lemma by showing that for any $i \in [n]$, based on $v(i)$, we can uniquely recover $v(i)$ and $v'(i)$.

Case 1: $\widetilde{v}(i) = \mathbf{0}$. It is obvious $v(i) = v'(i) = \mathbf{0}$.

Case 2: $\widetilde{v}(i) \notin V$ but $\widetilde{v} = cu$ for some $u \in V$ and some scalar $c$. In this case, $c$ must be either $s$ or $1 - s$. If $c = s$, then $v(i) = u, v'(i) = \mathbf{0}$. If $c = 1 - s$, then $v(i) = \mathbf{0}, v'(i) = u$.

Case 3: $\widetilde{v}(i) \notin V$ and $\widetilde{v} \neq cu$ for any $u \in V$ and any scalar $c \neq 0$. For any two $u, u' \in V$, let $\text{SPAN}(u, u')$ denote the vector space spanned $u$ and $u'$. Since $V$ is a linearly independent set, it is clear every choice of $(u, u')$ gives a different space $\text{SPAN}(u, u')$, and $\widetilde{v}(i)$ must live in one and only one such space. After identifying this space, we can identify $(u, u')$. With the knowledge of $s$, we can precisely correspond $u$ and $u'$ with $v(i)$ and $v'(i)$ since either $u = v(i)$ and $u' = v'(i)$ are true, or $u = v'(i)$ and $u = v(i)$ are true, but both can not be true at the same time.

Thus we have enumerated all possible cases, and in each case, there is a unique solution to the equation of interest. $\square$

## A.3 Proof of Lemma 3

*Proof:* Since the rows of $B$ are linearly independent, there is a unique $\widetilde{T}$ for which

$$
\widetilde{v} = \widetilde{T} B.
$$

We can recover $\widetilde{T}$ by projecting the rows of $\widetilde{v}$ on the basis $B$. It is clear $\widetilde{T} = sT + (1 - s)T'$. Then we only need to recover $T$ and $T'$ from $\widetilde{T}$. But since $\mathcal{T}$ is linearly independent and $T, T' \in \mathcal{T}$, following a similar argument as in Case 3 of the proof for Lemma 2, we see that $T$ and $T'$ can be uniquely recovered. $\square$.

### A.4 STATISTICS OF BENCHMARK DATASETS

Table 3 details the statistics of the 8 benchmark datasets used in the paper.

| Name | graphs | nodes | edges | features | classes |
|---|---|---|---|---|---|
| PTC_MR | 334 | 14.3 | 29.4 | 18 | 2 |
| NCI109 | 4127 | 29.7 | 64.3 | 38 | 2 |
| NCI1 | 4110 | 29.9 | 64.6 | 37 | 2 |
| MUTAG | 188 | 17.9 | 39.6 | 7 | 2 |
| ENZYMES | 600 | 32.6 | 124.3 | 3 | 6 |
| PROTEINS | 1113 | 39.1 | 145.6 | 3 | 2 |
| IMDB-M | 1500 | 13.0 | 65.9 | N/A | 3 |
| IMDB-B | 1000 | 19.8 | 96.5 | N/A | 2 |

Table 3: Statistics of the graph classification benchmark datasets.

### A.5 ILLUSTRATION OF MANIFOLD INTRUSION FROM MIXING GRAPH PAIRS

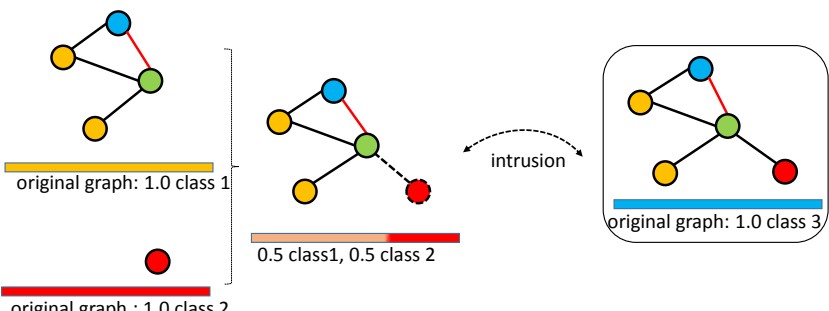

Figure 7: Manifold intrusion caused by connecting a graph pair. The synthetic graph in the middle is created by connecting the two graphs from the left, but assigning a soft label. This synthetic graph (with soft label) has the same structure as the right graph with one-hot label.

Figure 7 depicts an intrusion caused by connecting a graph pair. The synthetic graph in the middle is created by connecting the two graphs from the left, but assigning a soft label (i.e., 50% of class1 and 50% of class 2). This synthetic graph has the same structure as the right graph from the original training set, with an one-hot label (i.e., 100% of class 3).

### A.6 VISUALIZATION OF THE LEARNED REPRESENTATIONS

In Figure 8, we also visualize the final-layer representations formed by the GCN baseline, Mixup-Graph, and ifMixup on the original training graphs of the NCI109 and NCI1 datasets. We project these embeddings to 2D using t-SNE (van der Maaten & Hinton, 2008).

The upper row of Figure 8 shows that for NCI109, both GCN and MixupGraph were not able to separate the two classes, while ifMixup completely separated the training graphs with different labels. Similarly, when considering the bottom row of Figure 8 as for NCI1, both GCN and MixupGraph did not completely divide the two classes, while ifMixup attained a perfect separation for the two.

### A.7 UNEXPECTED RESULTS

We here also report an unexpected result.

We randomly shuffle the node order of one of the graphs in the graph pair before mixing for ifMixup. Such shuffling is able to significantly increase the number of synthetic graphs used for training for ifMixup, and we expect this would further improve the model's predicative accuracy. The accuracy

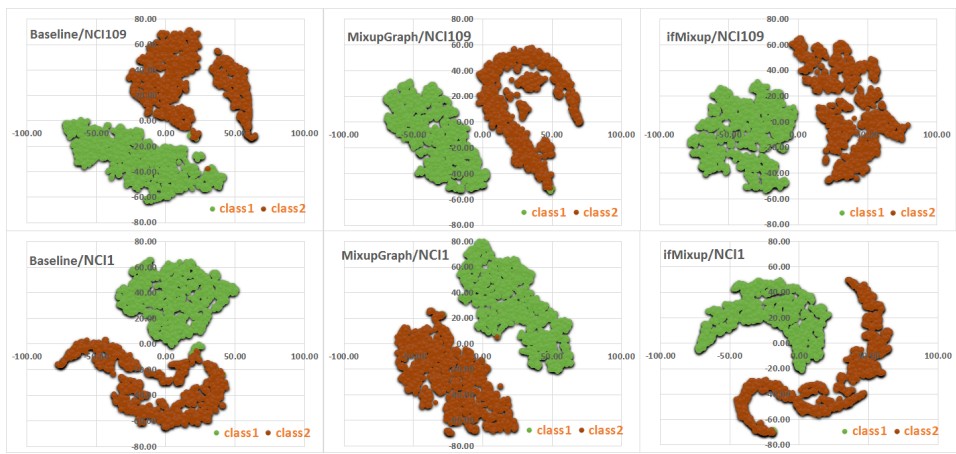

Figure 8: 2D visualization of the learned representations of the training graphs in NCI109 and NCI1.

obtained over the first six datasets of Table 3 obtained by ifMixup with GCNs as baseline is presented in Table 4.

Unexpectedly, results in Table 4 show that leveraging node order shuffling to increase the training data size did not help in terms of accuracy obtained. We hypothesis that this is may due to the modeling capability of the GCN networks. We will further investigate this hypothesis in our future work.

| ifMixup | without Shuffling | with Shuffling |
|---|---|---|
| PTC_MR | 0.654±0.003 | 0.650±0.004 |
| NCI109 | 0.820±0.005 | 0.816±0.001 |
| NCI1 | 0.819±0.004 | 0.817±0.001 |
| MUTAG | 0.879±0.003 | 0.864±0.006 |
| ENZYMES | 0.570±0.014 | 0.579±0.008 |
| PROTEINS | 0.753±0.008 | 0.741±0.003 |

Table 4: Accuracy of ifMixup with and without randomly shuffling the graph node order before mixing, with GCN networks as baseline.

## A.8 A VARIANT OF MIXUPGRAPH

For GIN, the final-layer representation of a graph is the concatenation of all the representations of each layer of the networks. In this sense, we can implement the idea of mixing on a random embedding layer as that in the Manifold Mixup (Verma et al., 2018) for vision.

We compare Manifold Mixup with the MixupGraph, and the results are shown in Table 5. Results in the table show that MixupGraph and Manifold Mixup obtained similar accuracy on all the eight datasets. For example, for the PTC_MR and IMDB-M datasets, Manifold Mixup obtained higher accuracy, while on the ENZYMES and NCI109 datasets Manifold Mixup obtained lower accuracy than MixupGraph. For the other four datasets, the accuracy obtained by the two methods are comparable.

|  | MixupGraph | Manifold Mixup |
|---|---|---|
| PTC_MR | 0.631±0.005 | 0.655±0.025 |
| NCI109 | 0.822±0.008 | 0.820±0.007 |
| NCI1 | 0.822±0.001 | 0.824±0.005 |
| MUTAG | 0.884±0.009 | 0.887±0.008 |
| ENZYMES | 0.521±0.007 | 0.505±0.028 |
| PROTEINS | 0.744±0.005 | 0.747±0.008 |
| IMDB-M | 0.518±0.004 | 0.521±0.002 |
| IMDB-B | 0.761±0.001 | 0.764±0.004 |

Table 5: Accuracy of the MixupGraph and Manifold Mixup with GIN networks as baseline. We report mean scores over 3 runs of 10-fold cross validation with standard deviations (denoted ±).

## A.9 SHALLOW GCN AND GIN

We also tested the performance of a 3-layer GCN and a 3-layer GIN baseline models. Results are presented in Table 6. As can be seen in the table, both GCN and GIN baselines obtained inferior

|  | GCN Baseline | GIN Baseline |
|---|---|---|
| PTC_MR | 0.619± 0.006 | 0.617± 0.003 |
| NCI109 | 0.791± 0.004 | 0.810± 0.002 |
| NCI1 | 0.796± 0.002 | 0.814± 0.001 |
| MUTAG | 0.827± 0.003 | 0.883± 0.009 |
| ENZYMES | 0.508± 0.015 | 0.497± 0.003 |
| PROTEINS | 0.738± 0.005 | 0.742± 0.007 |
| IMDB-M | 0.510± 0.008 | 0.511± 0.008 |
| IMDB-B | 0.748± 0.008 | 0.758± 0.002 |

Table 6: Accuracy of GCN and GIN with 3 layers. We report mean scores over 3 runs of 10-fold cross validation with standard deviations (denoted ±).

accuracy than a deeper GNN networks, namely 5 or 8 layers as used in the experiments in the main paper.

## A.10 ILLUSTRATION OF MIXING AND GRAPH STRUCTURE RECOVERING

In Figure 9, we illustrate how the mixed graph structure is formed and how the structures of the two source graphs can be recovered from the mixed graph.

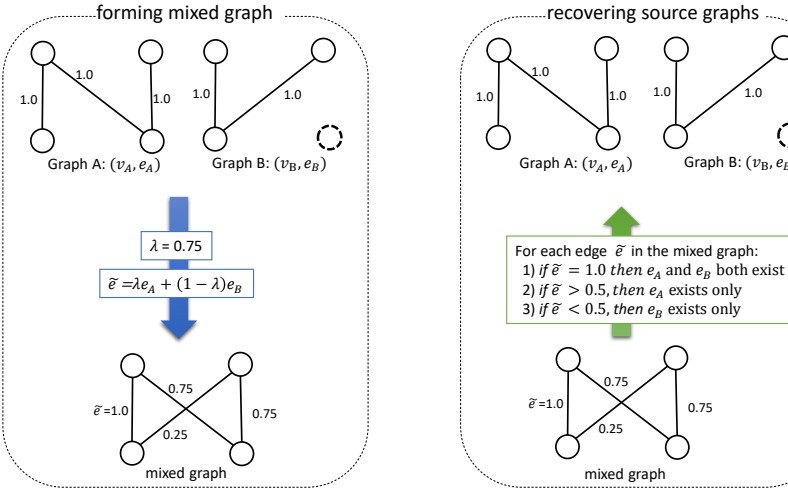

Figure 9: Illustration of mixing and recovering in ifMixup.

