# OpenReview forum: "Intrusion-Free Graph Mixup"
_ICLR.cc/2022/Conference — ICLR 2022 Submitted_

### Official Review · Reviewer_q8bs · 2021-11-02

**Correctness:** 3
**Technical Novelty And Significance:** 2
**Empirical Novelty And Significance:** 2
**Recommendation:** 5
**Confidence:** 4

**Main Review:**

Pros:
1. The research problem of data augmentation for graph classification is interesting and woth studying.
2. This paper is overall easy to follow.
3. The authors theoratically showed that the proposed ifMixup method is intrusion free.

Cons:
1. The performance improvements shown in Tables 2 and 3 are quite marginal.
2. The technical novelty is marginal considering the proposed ifMixup is basically a direct usage of Mixup on graphs.
3. From my point of view, one of the main challenges of applying Mixup on graphs is that the nodes in different graphs are not aligned. For example, the first feature of two different data objects indicates the same physical feature in regular data. However, the nodes in graph data are not nicely aligned like the features in regular data. Therefore, the way that the authors arbitrarily assign the node indeces (as stated in Sec3.2) is not very reasonable to me. I would appreciate if the authors can give more explaination and motivation on why they choose this random index and why this is a good solution.
4. Typos. e.g., A->B in Fig2; mixed usage of neighbor and neighbour.

**Summary Of The Paper:**

This work studied the research problem of data augmentation for supervised graph classification. The authors proposed ifMixup, which is a Mixup-based data augmentation method on graph data. The authors also showed that ifMixup is free of the known manifold intrusion problem of Mixup-based methods.

**Summary Of The Review:**

This work is very interesting, but I have slight concerns about part of the model design.

---

> ### Author Response · Authors · 2021-11-21
> **Response to Reviewer q8bs (1/2)**
>
> We would like to thank Reviewer q8bs for taking the time to review our paper. We appreciate  the insightful comments. We hope to address all concerns below.
>
> **[Q: The way that the authors arbitrarily assign the node indeces (as stated in Sec3.2) is not very reasonable to me.]**
>
> First we would like to clarify the classification setting of this work. Each input example is identified with a graph in which each node is identified with its feature (or embedding). Although each node in the graph is also given an ID, say an integer in {$1, 2, \ldots, n$} in a graph containing $n$ nodes, how the ID is assigned to each node is irrelevant to the class label of the input example. For such a classification task, all neural network models are also designed in a way that ignores the ID of the nodes while only exploiting the information about the connectivity between the node features. For this reason, the node ID assignment does not impact the performance of the classifier. On this basis, it is quite natural to use an arbitrary node ID assignment for graph A and an arbitrary node ID assignment for graph B and perform mixing of the two.
>
> But your question brings up a deeper question concerning the general methodology of Mixup-like training beyond graph Mixup considered in this paper. That is, in such training approaches, is it beneficial to perform mixing in a way that depends on the local semantics of two examples.  In the generic Mixup (Mixup: Beyond empirical risk minimization, ICLR2018.), such semantics information is not considered when perform mixing. For example, one may mix an outer space image and a water image, a tree image and a dog image; a patch of an image that corresponds to a dog may be mixed with a patch of another image that corresponds to the background or to a car.  This of course needs not to mean what images to mix and how to correspond the batches of one image with the batches of another image (or how to `"align'' the two images) in mixing is not an interesting direction that can bring fruitful discoveries and improvements -- Note that an image can be regarded as a special graph data instance in which the graph structure is a 2D lattice and each node features (in color image) is a three-dimensional vector (containing RGB values).
>
> Truly understanding what kind of graph alignment may bring more benefit requires  in-depth understanding of the working mechanism of Mixup itself. Before a better understanding is available in that direction, our semantics-ignorant alignment strategy seems the most natural choice. It is analogous to and as valid as what is being done in the generic Mixup for images, in which the alignment of local features is also semantics-ignorant.
>
> **[Q: The technical novelty is marginal considering the proposed ifMixup is basically a direct usage of Mixup on graphs.]**
>
> We would like to emphasize the fact that, until now no one has successfully introduced Mixup to graph inputs. The main challenge here is that graph-structured data have arbitrary structure and topology, as well as the manifold intrusion problem.  So applying Mixup to graph inputs are not trivial at all. This is also further justified by the reviewer's  comment "one of the main challenges of applying Mixup on graphs is that the nodes in different graphs are not aligned." In terms of novelty,  we are also the first to identify and discuss the manifold intrusion issue in graph data augmentation, and further show that our method can address this problem, which has not been discussed and coped with before in the graph learning field.

---

> > ### Author Response · Authors · 2021-11-21
> > **Response to Reviewer q8bs (2/2)**
> >
> > **[Q: The performance improvements shown in Tables 2 and 3 are quite marginal.]**
> >
> > We would like to emphasize the fact that these benchmark datasets are well-studied and well-tuned in the graph learning field, thus achieving a 1-2\% increase can be regarded as a remarkable improvement.
> >
> > Consider we compare Table 1 and Table 2. We can see that the much powerful GNN architecture GIN improves over the GCN architecture by mostly less than 2\%. For example on the NCI109 and NCI1 cases, the improvement of the current state-of-the-art (SOTA) GNN architecture GIN improves over GCN by only 1.7\% and 1.4\% respectively. On the other hand, our method improves over GIN by another 1.7\% and 2.1\% respectively, which in fact has established new SOTA accuracy against these two widely used benchmark datasets.
> >  Results in Tables 1 and 2 further show that our method can improve over the comparison baseline models (GCN and GIN)  by up to over 5\% (as shown in the last column of Table 1 and Table 2), which we think it is a very significant improvement. Also, if we compare our method with the widely used graph data augmentation method  DropEdge,  for example in the GIN case as shown in Table 2, our method improved over DropEdge by  4.5\%, 4.8\%, 3.6\%, and 5.5\% on NCI19, NCI1, MUTAG, and ENZYMES datasets, respectively, which indeed represent   large improvement margins.
> >
> > Furthermore, as shown in Table 1 and Table 2, all the other augmentation methods tested in the paper (namely MixupGraph, DropEdge, and DropNode) can degrade the predictive performance of the baseline models (for both GCN and GIN). Our method, however, never degraded the baseline models' predictive accuracy.
> >
> >
> > **[Q: Typos. e.g., A$->$B in Fig2; mixed usage of neighbor and neighbour.]**
> >
> > We have fixed the typo in Fig2 and consistently use "neighbor" in our paper. We appreciate the reviewer for reviewing our paper so carefully.

---

### Official Review · Reviewer_7hBS · 2021-11-03

**Correctness:** 2
**Technical Novelty And Significance:** 2
**Empirical Novelty And Significance:** 2
**Recommendation:** 5
**Confidence:** 4

**Main Review:**

Strengths
- the proposed mixup strategy is simple
- motivation for alleviating the manifold intrusion issue is clear
- I am satisfied with the parts on page 8 (the last sentence before section 4.2.3) and Figure 5, as the over-smoothing problem has been well-discussed. They demonstrate the proposed mixup method can work better when the over-smoothing issue is serious.
- eight graph classification benchmark datasets are used for evaluation.

Weaknesses
- in section 3.3, the authors proved the invertibility and concluded that the proposed method is manifold intrusion-free.
> 1) The proof of invertibility assumes the graph edge’s weight must be 0 or 1. Can the proposed method be used for the graph with real-value edges [0,1]?
> 2) When using a graph with real-value edge weights, does the proposed method still satisfy the invertibility?
> 3)The proof also requires the lambda not to be 0.5. But in standard mixup, the mixed sample can be generated in the case of lambda being 0.5. Thus, the assumption in this proof may not be correct.
> 4)The most important question in this part is how the invertibility can induce “manifold intrusion free”? The connection between them is unclear. Assume your mixed sample can recover two original interpolation graphs, how to ensure that there is no conflict between the mixed sample’s representation and other training samples’ ones. Please explain this connection.
- As you used 5- and 8-layer GNNs in experiments, why not consider shallower GNNs? As seen in Figure 5, baselines show better performance when using a shallower structure.
- it is unclear what do you want to prove in section 4.3.1 as there are no connections to the proposed method.
- there are several existing related papers that all use mixup for graph node-classification tasks. As your proposed method can be used for mixing two graphs, it can also mix one graph with itself. Besides, as in your title, it should be a general mixup method for graphs, right? For these reasons, it will be good to also do the evaluation on node-classification tasks.
Then SOTA graph mixup method should be [1]. I would like to see a comparison between the proposed method and baseline [1]. ([2] should be a weak baseline as it does not work significantly better than the basic baseline GNN, see Table3.)
[1] 2021AAAI GraphMix: Improved Training of GNNs for Semi-Supervised Learning
[2]2021 WWW MixupGrpah: Mixup for node and graph classification

**Summary Of The Paper:**

This manuscript develops a mixup strategy for graphs. Specifically, as claimed by the authors, the proposed method is manifold intrusion-free. This is achieved by mixing two input graph pairs (some dummy nodes are needed, see Figure2) based on the standard mixup formulation.

**Summary Of The Review:**

Overall, the proposed method is simple. Its main limitation is their theory on invertibility and their experiments. In their experiments, as a general data augmentation method for graphs, node-classification tasks are also needed. The baseline of MixupGrpah is a bit weak. Stronger baseline - GraphMix is encouraged to be used for comparison.

---

> ### Author Response · Authors · 2021-11-21
> **Response to Reviewer 7hBS (1/2)**
>
> We would like to thank Reviewer 7hBS for taking the time to review our paper. We appreciate  the insightful comments. Also, we would like to thank the reviewer  for highlighting  the capability of our method on coping with the over-smoothing issue, which is an extremely important but very challenge problem in graph learning.
>
> **[Q: Its main limitation is their theory on invertibility]**
>
> We truly appreciate  the reviewer for combing through our paper and raising the concern on our theory of  invertibility. We believe our revision has addressed the invertibility by adding Lemma 2, Lemma 3, and Theorem 1 in Section 3.3.
>
>  The newly added Lemma 2 and Lemma 3 guarantee that the node feature vectors of the two original graphs can be uniquely recovered from the mixed graph. These two lemmas, along with Lemma 1,   ensure that both the graph structures and the node features  of the two source graphs can be uniquely recovered from the mixed graph, resulting in the invertibility property of our approach.
>
>  Also, the newly added  Theorem 1  precisely proves that the invertibility  of our method guarantees that, for a mixed graph
>  there is no other graph pair  from the training set  that can be mixed into the same mixed graph using any mixing ratio $\lambda$.  Thus, manifold intrusion does not occur in ifMixup.
>
> **[Q: How the invertibility can induce “manifold intrusion free”?]**
>
> Our apologies for missing the connection between the invertibility and the manifold intrusion free. In a nutshell, manifold intrusion refers to that the data augmentation process  creates  graphs with identical structure but with  different labels. We precisely prove the manifold intrusion free property in our method by adding a new  theorem (i.e.,  Theorem 1) and  its proof in  Section 3.3.  This theorem proves that the invertibility  of our method guarantees that, for a mixed graph  there is no other graph pair  from the training set  that can be mixed into the same mixed graph using any mixing ratio $\lambda$.  Thus, manifold intrusion does not occur in ifMixup.
>
>  In addition, we also added two figures with concrete mixing ratio (in  Appendix: A.10-Figure 9) to illustrate the graph structure mixing and recovering processes of our method.
>
> **[Q: Can the proposed method be used for the graph with real-value edges [0,1]? does the proposed method still satisfy the invertibility?]**
>
> Our ifMixup scheme relies on the fact that input graphs do not have soft (weighted) edges, which is a widely adopted  assumption in the graph learning field.  At the end of this Section 3.3, we have added a detailed discussion on the potential of extending our mixing  scheme to  input graphs with  soft (weighted) edges.
>
> **[Q: In standard mixup, the mixed sample can be generated in the case of lambda being 0.5. Thus, the assumption in this proof may not be correct.]**
>
> We believe that our assumption is correct. In standard Mixup, the mixing ratio $\lambda$ is sampled from a standard Beta distribution, which is a continue distribution between (0, 1). As stated in the paper (under Lemma 1) "if $\lambda$ is drawn from a continuous distribution over (0; 1), the probability it takes value 0.5 is zero." In other words, the possibility of sampling a value of exact $\lambda$ = 0.5 from a continue distribution in (0, 1) for a Mixup method is in fact zero. So, our proof is correct.
>
> Intuitively, take any Beta distribution from Figure 3 of the paper. We can see that there exist countless points (i.e., $\lambda$ values) on the distribution curve, and 0.5 is just one of such countless points, and as such the chance of getting exact 0.5 by randomly picking a point from the countless points on the curve is thus zero.
>
> **[Q: Why not consider shallower GNNs? As seen in Figure 5, baselines show better performance when using a shallower structure.]**
>
> As suggested, we tested the performance of a 3-layer GCN and a 3-layer GIN  baseline models.
> Results are presented in Table 6. As can be seen in Table 6 in the Appendix, both GCN  and GIN baselines obtained inferior accuracy than a  deeper GNN networks, namely 5 or 8 layers as used in the experiments in the paper.

---

> > ### Author Response · Authors · 2021-11-21
> > **Response to Reviewer 7hBS (2/2)**
> >
> > **[Q: What do you want to prove in section 4.3.1]**
> >
> > In Section 4.3.1., the Manifold Mixup method can be considered as a variant of the MixupGraph. That is, the Manifold Mixup method reported in the paper  extends the MixupGraph with more flexible setting that enables generating more synthetic graphs for training. This is because Manifold Mixup method randomly samples a layer to conduct MixupGraph. We expected the Manifold Mixup method would obtain better results than MixupGraph. So the aim of Section 4.3.1. is to see if we could improve the performance of MixupGraph, the only available Mixup method for graph-level classification.
> >
> > To improve the paper's coherence, We have moved this section to the Appendix (i.e., A.8), and changed the title of the section as "A Variant of MixupGraph'' .
> >
> > **[Q: It will be good to also do the evaluation on node-classification tasks.]**
> >
> >  We agree with the reviewer that there are many research topics deserving pursuing in graph such as node classification, link prediction, graph classification etc. Our paper here aims at graph level classification. We will leave other directions such as node classification and link prediction in our future work.
> >
> > On the other hand, theoretically our ifMixup idea can be applied to node classification task, but some careful design considerations are needed to ensure that such Mixup can achieve better predictive accuracy than the baseline models. Since such design issues are not trivial,  we would like to leave such node classification task for future work.
> >
> > **[Q: The baseline of MixupGraph is a bit weak. Stronger baseline - GraphMix is encouraged to be used for comparison.]**
> >
> > In our humble view, GraphMix is designed for semi-supervised node classification task, as stated in the paper.
> > The method trains a fullyconnected network (FCN) jointly with the graph neural network, and the predictions made by the GNN for unlabeled data are used to augment the input data for the FCN. In this sense, graph node classification in a semi-supervised setting that leverages unlabeled graph nodes  is very  different from the setting and the main goal of our paper: graph level classification. Much efforts will be needed to modify the  GraphMix  method to a fully supervised graph level classification setting. We, therefore, believe comparing with that method is out-of-the-scope of this paper.
> >
> > To the best of our knowledge, the only available Mixup method for graph level classification is MixupGraph. That is the reason we conducted extensive comparison with this method. Besides, for better comparison we also constructed a variant of MixupGraph, namely Manifold Mixup, which generates more synthetic graphs for training (by randomly sampling a layer to perform MixupGraph) than the original MixupGraph.

---

> ### Author Response · Authors · 2021-11-22
> **Did we address your concerns on our work's limitation?**
>
> Dear Reviewer 7hBS,
>
> Thank you again for your constructive feedback.
> We have submitted very detailed responses to address your initial review comments. Are there any other concerns we didn’t address?

---

### Official Review · Reviewer_n1Dk · 2021-11-03

**Correctness:** 2
**Technical Novelty And Significance:** 2
**Empirical Novelty And Significance:** 2
**Recommendation:** 3
**Confidence:** 5

**Main Review:**

Strength:

1. Extending Mixup for Graph structured data is an important open problem and this paper attempts to address this.
2. Paper is written clearly at most places, however the mathematical notation in Section 3.3 can be simplified. This section is presenting a very simple argument which can be given without using so much mathematical notations.

Weakness:

The main claim of the paper that the proposed method avoids Manifold Intrusion, is unsubstantiated. It is correct that the edge connectivity of the individual graphs can be recovered from the mixed graph if the edges are unweighted, but how does it mean that the mixed graph  will not collide with some other existing graph or some other mixed graph. In the words of the authors "By this lemma, we see that in ifMixup, the two source graphs can be uniquely recovered from the mixed graph, without the knowledge of the mixing coefficient. Thus it is impossible for the graph resulting from mixing to coincide with any other graph in the training set or with a mixed graph from any other graph pairs" This is the main claim of the paper but there is no attempt to explain this."

Unless, I am missing something very trivial, the above claim is not straightforward to deduce from the fact that the edge connectivity of the individual graphs can be recovered from the mixed graph. Moreover, the edge connectivity can not be recovered from the mixed graph if the edges are weighted. Similarly the node features of the individual graphs can not be recovered from the mixed graph. If the authors can address this in the rebuttal, I am willing to increase the score.

The results are marginally better than the baseline that does the mixing in the representation space, and other baselines such as DropNode and DropEdge.


**Summary Of The Paper:**

The paper proposes a mixing strategy for graph structured samples. This strategy is works as follows : 1) take a pair of graphs 2) assign a node index from 1 to N to each node in the graphs, where N is the number of nodes in the larger graph. 3) for the smaller graph, create dummy nodes so that the total number of nodes in smaller graph becomes N. 4) Mix (interpolate) the node features and edge connectivity of the nodes with same indexes from the two graphs.

The main claim of this paper is that, such kind of mixing avoids the "manifold intrusion" problem.

**Summary Of The Review:**

The paper proposes a mixing strategy for the graph structured datasets, but the main claim of the paper that the method is "Intrusion Free" is not correct in my opinion.

---

> ### Author Response · Authors · 2021-11-21
> **Response to Reviewer n1Dk**
>
> We would like to thank Reviewer n1Dk for taking the time to combing through our paper so carefully.  We truly appreciate  the reviewer's insightful comments, in particular raising the node feature question  regarding our proof on invertibility. We hope to address all concerns below.
>
>  **[Q: The node features of the individual graphs can not be recovered from the mixed graph].}**
>
>  We would like to thank the reviewer for raising this question.  Two new lemmas (i.e., Lemma 2 and Lemma 3), along with their proofs, have been added to Section 3.3 to address the inevitability of node feature vectors. The newly added Lemma 2 and Lemma 3 guarantee that the node feature vectors of the two original graphs
> can be uniquely recovered from the mixed graph.
>
> These lemmas, along with Lemma 1,  ensure that both the graph structures and the node features  of the two source graphs can be uniquely recovered from the mixed graph, resulting in the invertibility property of our approach.
>
> **[Q: How does it mean that the mixed graph will not collide with some other existing graph or some other mixed graph.]**
>
> Our apologies for missing the connection between the invertibility and the manifold intrusion free. In a nutshell, manifold intrusion refers to that the data augmentation process  creates
>  graphs with identical structure but with  different labels. We precisely prove the manifold intrusion free property in our method by adding a new  theorem (i.e.,  Theorem 1) and  its proof in  Section 3.3.  This theorem proves that the invertibility  of our method guarantees that, for a mixed graph  there is no other graph pair  from the training set  that can be mixed into the same mixed graph using any mixing ratio $\lambda$.  Thus, manifold intrusion does not occur in our method ifMixup.  In addition, we also added two figures with concrete mixing ratio (in  Appendix: A.10-Figure 9) to illustrate the graph structure mixing and recovering processes of our method.
>
> **[Q: The main claim of the paper that the method is "Intrusion Free" is not correct in my opinion.]**
>
> We would like to thank the reviewer for  raising the questions of invertibility of node features  and  the missing of  connection between invertibility and manifold intrusion free. We believe that we have addressed this main claim concern by answering the above two questions.
>
> **[Q: The edge connectivity can not be recovered from the mixed graph if the edges are weighted]**
>
> Our ifMixup scheme relies on the fact that input graphs do not have soft (weighted) edges, which is a widely adopted  assumption in the graph learning field.
>  At the end of Section 3.3, we have added a detailed discussion on the potential of extending our mixing  scheme to  input graphs with  soft (weighted) edges.
>
> **[Q: The results are marginally better than the baseline]**
>
> We would like to emphasize the fact that these benchmark datasets are well-studied and well-tuned in the graph learning field, thus achieving a 1-2\% increase can be regarded as a remarkable improvement.
>
> Consider we compare Table 1 and Table 2. We can see that the much powerful GNN architecture GIN improves over the GCN architecture by mostly less than 2\%. For example on the NCI109 and NCI1 cases, the improvement of the current state-of-the-art (SOTA) GNN architecture GIN improves over GCN by only 1.7\% and 1.4\% respectively. On the other hand, our method improves over GIN by another 1.7\% and 2.1\% respectively, which in fact has established new SOTA accuracy against these two widely used benchmark datasets. Results in Tables 1 and 2 further show that our method can improve over the comparison baseline models (GCN and GIN)  by up to over 5\% (as shown in the last column of Table 1 and Table 2), which we think it is a very significant improvement. Also, if we compare our method with the widely used graph data augmentation method  DropEdge,  for example in the GIN case as shown in Table 2, our method improved over DropEdge by  4.5\%, 4.8\%, 3.6\%, and 5.5\% on NCI19, NCI1, MUTAG, and ENZYMES datasets, respectively, which indeed represent   large improvement margins.
>
> Furthermore, as shown in Table 1 and Table 2, all the other augmentation methods tested in the paper (namely MixupGraph, DropEdge, and DropNode) can degrade the predictive performance of the baseline models (for both GCN and GIN). Our method, however, never degraded the baseline models' predictive accuracy.
>
> **[Q. The mathematical notation in Section 3.3 can be simplified]**
>
> We thank the reviewer for the suggestion of simplifying the notations in Section 3.3.  We have restructured the entire Section 3 to make the notations and text much clearer than the original version.

---

> ### Author Response · Authors · 2021-11-22
> **Are there any other concerns we didn’t address?**
>
> Thank you again for your constructive feedback. We have submitted very detailed responses to address your initial review comments. Are there any other concerns we didn’t address?

---

### Official Review · Reviewer_d3Ri · 2021-11-05

**Correctness:** 4
**Technical Novelty And Significance:** 3
**Empirical Novelty And Significance:** 3
**Recommendation:** 8
**Confidence:** 4

**Main Review:**

This paper proposes a very simple and logical algorithm for performing input mixup on graphs, and achieves excellent results with this technique across a wide variety of datasets.  It is also shown that this logically allows one to gain improvements from larger models and also from larger mixing rates.  It suggests many natural and interesting follow-ups, such as doing this same mixing in random layers, yet I feel this is a healthy and useful contribution for the field.

Small Comments:
  -Improvements are pretty nice and the table is very readable!
  -Additionally, improvement's stability as we increase mixing rate is nice to see.



**Summary Of The Paper:**

This paper shows a very simple technique for performing input-space mixup on graphs, in which the vertices are put into an arbitrary order and dummy vertices are added to make the two graphs being mixed have the same number of vertices.  Mixing is then performed on the vertices and edges.  The improvements are excellent and seem to be SOTA on all benchmarks considered, often by nice margins.

Detailed Notes from reading paper:
  -Mixing on graphs.  Theoretical guarantee that mixed graphs avoid intrusions.
  -Example of intrusion on mixed graphs is interesting, simple, and logical.
  -Proposed algorithm is "Intrusion-Free Mixup" ifMixup.
  -Edges and nodes take interpolated values.  Labels are interpolated based on edges.
  -Can recover source graphs from mixed graphs.
  -Task is to convert each graph into a single class (as opposed to each node having a class label).
  -Each node is assigned a distinct index, dummy nodes are created to make each "graph" same size.  Dummy nodes have no edges.
  -First construct "soft" edges by mixing edges between two graphs.  Network needs to be able to take soft edges.
  -Proof that this graph mixing operation is invertible.
  -ifMixup also shows benefit from more depth, as compared to baseline (which is probably badly overfitting).


**Summary Of The Review:**

The paper shows a way of interpolating graphs for graph classification and achieves excellent empirical results while showing that the interpolated graphs are invertible functions of the original graphs, avoiding the underfitting issues that may occur with other techniques for applying input mixup to graphs.

---

> ### Author Response · Authors · 2021-11-21
> **Response to Reviewer d3Ri**
>
> We would like to thank Reviewer d3Ri for taking the time to review our paper so thoroughly.
> We truly appreciate  the reviewer's insightful comments and thank the reviewer for the kind words!

---

### Author Response · Authors · 2021-11-21
**Updates in  Revision**

We truly appreciate  the reviewers for their many helpful comments. We have revised our paper to hopefully address all of their concerns. We summarize  the main changes of our paper as below for convenience.

 1. Two new lemmas (i.e., Lemma 2 and Lemma 3), along with their proofs, have been added to Section 3.3. In a nutshell, Lemma 1 ensures that the graph structures of the original two graphs can be uniquely recovered from the
mixed graph. The newly added Lemma 2 and Lemma 3 guarantee that the node feature vectors of the two original graphs
can be uniquely
recovered from the mixed graph.
These lemmas  ensure that both the graph structures and the node features  of the two source graphs can be uniquely recovered from the mixed graph, resulting in the invertibility property of our approach.   We would also like to reiterate how grateful we are to reviewer n1Dk  for raising the   node features invertibility question.

 2. A new  theorem (i.e.,  Theorem 1), along with its proof, has been added to Section 3.3.  This theorem precisely proves that the invertibility  of our method guarantees that, for a mixed graph
 there is no other graph pair  from the training set  that can be mixed into the same mixed graph using any mixing ratio $\lambda$.  Thus, manifold intrusion does not occur in our method ifMixup.
 We hope the newly added Theorem 1 has addressed the main concern of Reviewer n1Dk and Reviewer 7hBS, namely the missing of the connection between our model's  invertibility property and the manifold intrusion free guarantee.
 In addition, we also added two figures with concrete mixing ratio (in  Appendix: A.10-Figure 9) to illustrate the graph structure mixing and recovering processes of our method.

 3. We re-organized the entire Section 3 to make the notations and text much clearer than the original version.
Further more, at the end of Section 3.3, we have added a detailed discussion on
 extending our mixing  scheme to  input graphs with  soft (weighted) edges.

 4. Additional experiments using shallow GCN and GIN baselines  have been added to the experiments (in Appendix: A.9-Table 6).

5. We made the novelty of our paper more specific, including the follows.
 We are the first to identify and discuss the manifold intrusion problem in graph data augmentation. Also,  it is not trivial to apply Mixup to  graph inputs, and we are the first to  successfully propose such method.  More promisingly, our method  is manifold intrusion free, and obtained superior accuracy  over several comparison baseline models (with large margins in some cases). Moreover,  unlike all the other baseline models we tested, our methods never degraded the baseline models' predictive accuracy.

6. All typos, unclear explanations,  and redundant notations have been fixed. We hope they are much clearer now.

In summary, by added Lemma 2, Lemma 3, and Theorem 1, we hope that all the reviewers' main concerns have been addressed. We would also like to  reiterate our appreciation for their feedback, which has significantly improved our paper.

---

> ### Comment · Reviewer_q8bs · 2021-11-21
> **request for coloring changes**
>
> Just a quick note. I would appreciate if the authors can highlight the changes with colored text so we can easily find the updates.

---

> > ### Author Response · Authors · 2021-11-21
> > **colored text version uploaded**
> >
> > Thank you for the suggestion. We have uploaded a version with added contents (text, tables, and figures) colored in blue (in both the main paper and the appendix).

---

### Author Response · Authors · 2021-11-29
**Any questions before the end of the discussion period?**

We would like to thank again all reviewers.

Please let us know if there are additional questions or concerns before the end of the discussion period.
We would be happy to discuss or address any additional comments.

---

### Decision · Program_Chairs · 2022-01-20

**Decision:**

Reject

**Comment:**

This paper proposes a “Mixup” type of data augmentation for graphs that accounts for the difficulty of mixing graphs of different number of nodes. The authors show that the mixed graphs are invertible functions of the original graphs.

Reviewer d3Ri liked the simplicity and effectiveness of the technique. They called it a “healthy and useful contribution for the field”. Reviewer n1Dk thought that the paper explored an important problem and thought the paper was clear, though some of the math could have been simplified. This reviewer was concerned that a central claim of the paper, that the method avoids “Manifold Intrusion” was unsubstantiated. Specifically that it could not be deduced from the fact that edge connectivity could be recovered from the mixed graphs. The reviewer claimed that node features of the individual graphs were unrecoverable. The authors responded in detail to the reviewer’s criticism, adding two new lemmas which purportedly guaranteed node feature vectors could be uniquely recovered. The authors admitted to some conversion between “Manifold intrusion” and invertibility and added a Theorem and its proof that invertibility guarantees no manifold intrusion. The authors also responded to reviewer n1Dk’s concern about the significance of the reported improvements. Reviewer n1Dk responded to the author rebuttal with concerns about the strong and unrealistic assumption of linear independence of the feature matrix. They had further concerns that for the case of weighted edges the “Intrusion-free” property could not be enforced. Discussion ensued, with the authors arguing that the independence assumption was not as strong as the reviewer claimed and that the “Intrusion-free” property was only every for graphs with binary edge weights.

Reviewer 7hBS and q8bs were both on the fence. 7hBS also raised some concern with the case of non-binary weighted edges. They also raised the same issue with respect to the connection b/w invertibility and the “Intrusion-free” property, which the authors addressed. Reviewer q8bs also thought the problem was interesting, the paper was clear, yet like n1Dk thought the performance improvement was marginal and had concerns with technical novelty of the work.

This was a tough call, so I engaged the reviewers in further discussion. 7hBS agreed with n1Dk’s opinion that the central claim of the paper (the method being intrusion-free) was not presented with strong evidence. They also raised another concern, which was that the paper didn’t evaluate on node classification like most other graph mixup-style models. Q8bs agreed with n1Dk’s concerns and felt that post discussion the technical novelty of the work was limited. Without strong support from the reviewers, I think that this paper could use further development, either lightening the “intrusion-free” claim or presenting evidence for it in other settings.